# Hydrogen Production from Methanol–Water Solution and Pure Water Electrolysis Using Nanocomposite Perfluorinated Sulfocationic Membranes Modified by Polyaniline

**DOI:** 10.3390/polym14214500

**Published:** 2022-10-24

**Authors:** Carlos Sanchez, Francisco J. Espinos, Arturo Barjola, Jorge Escorihuela, Vicente Compañ

**Affiliations:** 1Instituto de Ingeniería Energética, Universitat Politècnica de València, C/Camino de Vera s/n, 46020 Valencia, Spain; 2Centro de Investigación en Acuicultura y Medio Ambiente (ACUMA), Universitat Politècnica de València, Campus de Vera s/n, 46020 Valencia, Spain; 3Departamento de Termodinámica Aplicada, Escuela Técnica Superior de Ingenieros Industriales (ETSII), Universitat Politècnica de València, C/Camino de Vera s/n, 46020 Valencia, Spain; 4Departamento de Química Orgánica, Universitat de València, Avda, Vicente Andrés Estellés s/n, 46100 Valencia, Spain

**Keywords:** water electrolysis, methanol electrolysis, perfluorinated sulfocationic membranes, polyaniline, PEMWE, hydrogen production

## Abstract

In this work, we report the preparation of Nafion membranes containing two different nanocomposite MF-4SC membranes, modified with polyaniline (PANI) by the casting method through two different polyaniline infiltration procedures. These membranes were evaluated as a polymer electrolyte membrane for water electrolysis. Operating conditions were optimized in terms of current density, stability, and methanol concentration. A study was made on the effects on the cell performance of various parameters, such as methanol concentration, water, and cell voltage. The energy required for pure water electrolysis was analyzed at different temperatures for the different membranes. Our experiments showed that PEM electrolyzers provide hydrogen production of 30 mL/min, working at 160 mA/cm^2^. Our composite PANI membranes showed an improved behavior over pristine perfluorinated sulfocationic membranes (around 20% reduction in specific energy). Methanol–water electrolysis required considerably less (around 65%) electrical power than water electrolysis. The results provided the main characteristics of aqueous methanol electrolysis, in which the power consumption is 2.34 kW h/kg of hydrogen at current densities higher than 0.5 A/cm^2^. This value is ~20-fold times lower than the electrical energy required to produce 1 kg of hydrogen by water electrolysis.

## 1. Introduction

Hydrogen is considered one of the most important energy carriers, since it is the most efficient alternative to conventional fossil fuels, and constitutes a clean and sustainable source of energy due to its great abundance on the Earth’s surface. The current state of fossil fuels combined with increasing energy demand has turned the scientific community’s attention toward sustainable processes to obtain hydrogen for its use as an “energy vector”, due to its high energy density (140 MJ/kg, almost three times higher than that of typical solid fuels) [1]. Hydrogen can be produced by five different procedures such as electrochemical, thermochemical, photochemical, and biological processes and electrolysis [2,3,4,5,6,7]. Despite each method having its advantages and drawbacks, the most sustainable methodology for pure hydrogen production on a small scale is polymer electrolyte membrane water electrolysis (PEMWE) [8,9]. In typical PEMWE cells, a proton electrolyte membrane (PEM), which is a semipermeable membrane acting as an electrolyte with the capacity to conduct protons, constitutes the central core of this electrochemical system [10,11]. Despite water electrolysis using an alkaline electrolyte and nickel electrodes having been widely used in many commercial applications [12], the use of polymers as acidic electrolytes has many advantages in comparison with water–alkaline electrolyzers, such as environmental cleanliness, compactness, high power density at relatively low operating temperatures (below 100 °C), and higher hydrogen-production rates. Among the different polymers used in PEMWE, perfluorosulfonic acid solid electrolytes, such as Nafion-based membranes, have demonstrated their potential for this application despite its high cost. In this regard, the operating cost of PEMWEs can be reduced by developing electrolyzers that operate at high current densities without the need to increase the operating voltage. To achieve this goal, the improvement of membranes with reduced cost and enhanced conductivity is fundamental. In this scenario, composite Nafion membranes can be fabricated in combination with other less expensive conductive polymers such as polyaniline (PANI) and polypyrrole (PPY) [13,14].

The production of hydrogen through water electrolysis is, thus, currently seen as a promising option for storing electrical energy from renewable energy sources such as wind and solar power and is also an interesting mechanism for decarbonizing transport and industry [10]. In the coming decades, water electrolysis will become the primary source of energy to produce hydrogen for use in the mobility sector via fuel cells, both in households and as the raw material for heavy industries [15]. Water electrolysis, so far, has not had a significant commercial impact due to its high energy requirements (4.5–5 kWh/Nm^3^ of H_2_) and low hydrogen evolution rate, which means it has not been cost-effective, even though it is a simple process and delivers clean gas. Electricity is known to be the most expensive form of energy, and hydrogen production efficiency is currently too low to be economically viable [12,13,14,15,16,17]. The global worldwide hydrogen production is around 500 billon cubic meters (b m^3^) per year, which is only 4% of the global industrial hydrogen produced by water electrolysis [17,18,19,20,21]. The growing interest in producing hydrogen by sustainable procedures has focused attention on PEMWE [22]. Alternative systems have also been developed and are based on methanol–water solutions to generate hydrogen, as a low operating voltage (about 0.02 V) is required compared with PEMWE (around 1.4 V below standard conditions). However, methanol-assisted water electrolysis (MAWE) has the drawback of producing carbon dioxide as a byproduct [19].

Recent studies carried out on the use of different elements in methanol–water electrolysis [19] describe the use of different types of catalysts (PtRu, Pt/C-SnO_2_, Pt/C-CeO_2_, and others) [20,21]. Recently, Pt was used on top of conventional PtRu for the electrolysis of ethanol and methanol. The methanol concentrations tested the range from 1 to 20 M. When the concentration is higher, the current density required is lower due to the partial dissolution of the Nafion membrane, which increases its conductivity. However, not only different catalysts are being used but also new membranes, such as PVDF/ZrP [20], which has better performance than Nafion when a methanol solution is electrolyzed in water, so the membrane is impermeable to the solution. In conventional PEMWE, the active surface of the membrane electrode assembly (MEA) is obtained with metallic platinum as electrocatalyst. In order to reduce the costs in the MEA construction, a Pt/C electrocatalyst has been used for the hydrogen evolution reaction (HER), and iridium dioxide (IrO_2_) has been used for the oxygen evolution reaction (OER) as the catalyst. This kind of catalyst has been used previously due to the highly acid environment found in proton-conducting solid polymer electrolytes [23]. Methanol is also envisioned as an energy vector to enable long-distance hydrogen transport. When the destination is more than 7000 km away, generating synthetic methanol from hydrogen and then electrolyzing it at the destination is more economically viable than transporting liquid hydrogen [24]. Electrolysis of a methanol solution has advantages when implementing integrated solutions, such as small-scale hydrogen generation using solar photovoltaics [25].

In this work, we report on the study of a Nafion-based composite membrane modified with nanocomposite PANI membrane, which have been shown to enhance fuel cell performance. The use of PANI fillers in the MF-4SC membranes presumably causes percolation pathways, which facilitate proton diffusion takes and, consequently, favor hydrogen generation. The results of hydrogen production by two different methods: are (1) methanol–water solution electrolysis at different methanol concentrations by means of an electrolytic cell and a Nafion 115 membrane; (2) PEM water electrolysis using a perfluorinated sulfocationic membrane (MF-4SC) and two composite membranes (MF-4SC/PANI) prepared under different oxidation times of aniline. For both methods, we used the low-cost hydrogen evolution reaction (HER) as the cathode electrocatalyst, and iridium oxide (IrO_2_) on the surface was used for the oxygen evolution reaction (OER), as the anode at different temperatures, to obtain the characteristic I-V, P-I curve, which is used to describe systems of hydrogen production.

## 2. Materials and Methods

### 2.1. Membrane Preparation

A commercial 20 wt% Nafion dispersion in water (DuPont Co., Wilmington, DE, USA) was mixed in isopropanol to prepare a 5 wt% solution in isopropanol and water (with isopropanol:water = 4:1, *w*/*w*), and membranes were prepared by traditional casting method. This solvent ratio has previously been reported to be suitable for Nafion infiltration through porous membranes [26]. Extra pure isopropanol and cetyltrimethylammonium bromide (CTAB) were purchased from Acros Organics (Fisher Scientific SL, Madrid, Spain) and *4-formyl-1,3*-benzenedisulfonic acid disodium salt from Sigma-Aldrich (Sigma-Aldrich Química SL, Madrid, Spain). The Nafion membranes were annealed at 125 °C for 90 min and then removed from the Petri dish by adding water. The last step was conditioning the membrane by treatment with water at 85 °C for 30 min, followed by wetting with a 3 wt% hydrogen peroxide solution for 1 h at 80 °C and further protonation at the same temperature by ion-exchange with a 1 M chlorohydric acid solution for 1 h. The Nafion membranes were then washed in hot water at 85 °C and finally dried, yielding membranes of 155 μm thickness. A perfluorinated sulfocationic membrane (MF-4SC) from Plastpolymer (Plastpolymer Joint Stock Company, St. Petersburg, Russia) with specific properties, such as high thermal and chemical stability, along with good protonic conductivity, was used as a template for composite membrane preparation. Two composite membranes were obtained from MF-4SC through Polyaniline (PANI) penetration into the initial polymer matrix of the membrane. The oxidative thermal conditioning of the membrane consisted of sequential boiling of MF-4SC membranes in 5% HNO_3_, 10% H_2_O_2_ aqueous solutions, and distilled water, for 3 h in each case [27].

The template synthesis of polyaniline in the perfluorinated sulfocationic membrane was performed by an optimized chemical method, as previously described [28,29,30]. The membrane is sandwiched between two solutions, and different synthetic approaches were followed. In the first approach, the polymerization was carried out in a two-chamber cell by the counter diffusion method. In this methodology, a vertically fixed membrane is positioned between two different oxidizing solutions: the first of 0.01 M FeCl_3_ in a 0.5 M H_2_SO_4_ aqueous solution (24 h) and the second of 0.01 M aniline in a 0.5 M H_2_SO_4_ aqueous solution (24 h). Samples prepared by this methodology are labeled with an asterisk (*). These samples were saturated in a solution of 0.01 M An + 0.5 M H_2_SO_4_ for 24 h and then in a mixture of 0.01 M An + 0.01 M FeCl_3_ + 0.5 M H_2_SO_4_ solutions (3 h). In the other synthetic approach, the composite membranes were obtained by means of successive diffusions of the different solutions. The first step is performed with a 0.01 M aniline solution in 0.5 M H_2_SO_4_ (24 h), while the second step consisted of a 0.01 M FeCl_3_ solution in 0.5 M H_2_SO_4_. In both approximations, the membrane turned blue and then rapidly changed to emerald-green, after a period that depended on the sample pre-treatment. As a result, composite membranes (MF-4SC/PANI) with different color intensities could be prepared by varying the corresponding oxidation time. The color intensity is dependent on the exposure time in the oxidizing solutions [30]. Samples prepared by this methodology are labeled with a double asterisk (**). Table 1 shows the physical and chemical characterization of the prepared membranes. Saturation degree (%) was calculated as (g PANI/g wet membrane) × 100. Water uptake (%) is given as (g wet membrane/g dry membrane) × 100.

### 2.2. Membrane Electrode Preparation

The MF-4SC, Nafion 115, and nanocomposite PANI-membranes were prepared by dispersing the catalyst nanoparticles in the appropriate amounts of water (5% Nafion ionomer solution and isopropyl alcohol) for the tests on the electrolyzer. Pt (30 wt%) onto Vulcan carbon powder XC72, with a loading of 0.5 mg/cm^2^ on the surface, was used as electrocatalyst for the hydrogen evolution reaction (HER) as cathode, and 2 mg/cm^2^ of iridium oxide (IrO_2_) on the surface was used for the oxygen evolution reaction (OER) as anode. The membrane formed of both catalysts had an active surface area of 6.25 cm^2^. The assembly was hot-pressed at 5 MPa for 3 min at 120 °C.

### 2.3. Microscopy Measurements

The morphology and microstructure of the anisotropic composite membranes were characterized using scanning electron microscopy (SEM) recorded in the tapping mode by a Hitachi S-3200N. SEM transversal and longitudinal sections of the samples were dried and sputtered with gold to reduce charging effects during the measurements. The observations were made at 15 kV. The morphology of membranes is shown in Figure 1, where the cross section of an MF-4SC membrane (Figure 1c) and the anisotropic composite membrane MF-4SC/PANI-3H* (Figure 1d) display the presence of grains approximately 2–10 µm in size, with the fractal dimension of polyaniline particles of about 2.5 µm on the surface of the composite membranes.

### 2.4. Experimental Test System

An experimental test system was set up to characterize the membranes. Figure 2 shows a diagram of the connection of all the elements. The device under test was fed with a constant supply of different concentrations of the methanol solution by a JASCO PU-2080 Plus pump from the reservoir at the appropriate concentration for each stage. The water contained in the solution ensured that the membrane was adequately wet. The methanol solution flow rate was precisely measured by a Biotech VZS-005-VA flow meter. The stainless-steel transducer consisted of a small turbine that provided 1800 pulses/L in the range between 0.005 L/min and 1.75 L/min. The 15 V power supply was provided from “Card #2”, with the signal conditioning circuit.

The testing device consisted of a single cell in which the membrane, electrodes, and plates were sandwiched between two pieces of methacrylate with an active area of 6.25 cm^2^, to provide the appropriate mechanical consistency and seal the assembly. A K-type thermocouple was inserted between, in contact with the membrane, to measure the temperature inside the cell. The contact between two both bimetals of the thermocouple was Teflon-coated to ensure a good isolation. Figure 3 shows images of the cell with the thermocouple inserted. The thermocouple signal conditioning was controlled through “Card #1” by an AD595 integrated circuit. This device eliminated the error produced by the cold junction of the thermocouple with the copper contact and provided 10 mV/°C by a K-type thermocouple.

An operational amplifier was used to adapt the voltage to that required by the measuring system. Temperature was measured by a simple LM35 temperature semiconductor sensor in the ambient temperature range between −55 to 150 °C. The temperature of the methanol solution in the reservoir and the ambient temperature were assumed to be the same.

The current needed to electrolyze the methanol was supplied by a Promax FAC 363-B DC power supply, configured to provide a constant current over the appropriate range of voltages. Current was measured by “Card #1”, by means of an LEM LA-25NP/SP11, a Hall effect current sensor that provided a constant current proportional to the measured current, with a transformation ratio of 25:1000 mA. A 120 ohms precision resistor was connected to the output of the sensor to convert the current to voltage, while a high impedance amplifier was used to adapt the voltage. The hydrogen produced by the device was measured by an AALBORG GFM 17 hydrogen flow meter, calibrated by the manufacturer, with the information on the type of gas to be measured provided by the research team and a maximum flow rate of 30 mL/min. Considering hydrogen flammability, the generated hydrogen was consumed by a PEM fuel cell to generate the power needed to move a small electric motor to move a small fan. The water produced by the fuel cell was measured by a calibrated measuring cylinder.

The system used to characterize the membrane performance consisted of:Cell voltage, range: 0–10 V;Cell current, range: 0–50 A;Ambient temperature, range: 0–150 °C;Cell temperature, range: 0–150 °C;Methanol or pure water flow, range: 0.005–1.75 L/min;Hydrogen flow, range: 0–600 mL/min (gaseous hydrogen);Waste produced (water).

Two electronic boards were built to collect and condition all sensor signals prior to input into the data acquisition system. Some of the conditioning needed for the more special sensors has been described above, while only a voltage limitation was enough to protect the inputs of the data acquisition system in the remaining cases. Figure 4 shows an image of the two interface cards.

The data acquisition system was a Measurement Computing USB-1608G, which has 16-bit analog measurement channels, two pulse counters, and digital inputs. In our case, we used five analog channels (cell voltage, cell current, ambient temperature, cell temperature, and hydrogen flow) and a pulse counter (methanol flow). Software processing was by performed using National Instruments LabVIEW. Samples of the analog channels were set every 0.5 s, and the screen representation was set to 10 s.

### 2.5. Methanol–Water Solution Electrolysis

For the hydrogen production by methanol–water solution electrolysis, the solution at different methanol concentrations (1, 2, 3, and 4 M) was supplied to the anode of the electrolytic cell using a DC power supply. Methanol reacts with water at the anode to produce carbon dioxide, protons (*H^+^*), and electrons (*e^−^*), according to the anode reaction given by Equation (1):(1)$$CH_{3}OH+H_{2}O\left(liq\right)\to CO_{2}+6H^{+}+6e^{-}$$

The carbon dioxide produced at the anode is extracted outside the anode. The protons were diffused to the cathode through the protonic exchange membrane, and the electrons produced are also conducted to the cathode through the external circuit, which contains the DC power supply. The electrons and protons from the anodic reaction react to produce hydrogen at the cathode as shown by Equation (1) [31]: (2)$$6H^{+}+6e^{-}\to 3H_{2}$$

The full reaction produced in the methanol–water solution electrolysis combines both anode and cathode reactions in the cell, yielding carbon dioxide and hydrogen as final products, as shown by Equation (3):(3)$$CH_{3}OH+H_{2}O\left(liq\right)\to CO_{2}+3H_{2}\left(g\right)$$

The variation of enthalpy (Δ*H*^0^) and free Gibbs energy (Δ*G*^0^) associated with Equation (3) can be calculated from the energy of the formation of methanol, carbon dioxide, and water, which under standard conditions are Δ*H*^0^ = 131.3 kJ and Δ*G*^0^ = 9.3 kJ/mol per mol of *H*_2_, respectively. The standard potential at the anode for the methanol reduction is 0.016 V versus NHE [32].

### 2.6. Pure Water Electrolysis

To produce hydrogen in an electrolysis cell using pure water, splittingwater molecules into gaseous *H*_2_ and *O*_2_ is realized by applying an electric current to the cell. In a PEM water electrolyzer, hydrogen is produced by supplying water to the anode, where it is decomposed according to Equation (4):(4)$$2H_{2}O(liq)\to 4H^{+}+O_{2}+4e^{-}$$

The protons are transported through the membrane to the cathode. The electrons exit to the electrolysis cell via an external circuit, which supplies the cell potential for the reaction. At the cathode, the electrons and protons recombine to give *H*_2_ gas, as previously shown in Equation. Thus, the overall reaction in the electrolysis cell corresponding to the water dissociation reaction is:(5)$$H_{2}O(liq)\to H_{2}(g)+\frac{1}{2}O_{2}(g)$$

In equilibrium, the amount of electricity required to split one mole of water is given by Equation (6): (6)$$\Delta G=nFE_{rev}$$
when *G* > 0 is the free energy change associated with water dissociation, *n* is the number of electrons exchanged during electrochemical splitting of water (*n* = 2), F is the Faraday constant (*F* = 96485.3 C/mol), and *E_rev_* is the reversible thermodynamic voltage associated with the water dissociation reaction. Taking into account that activating Gibbs free energy of the reaction can be expressed as
(7)ΔG(T,p)=ΔH(T,p)−TΔS(T,p)>0
where ΔH(T,p) and ΔS(T,p) are, respectively, the enthalpy change and entropy change associated with the water reaction given by Equation (6). Notice that such changes are a function of the operating temperature and pressure. To split a mole of water, the thermodynamic voltage is given by Equations (6) and (7) as
(8)$$E_{rev}=\frac{\Delta H(T,p)}{nF}-\frac{T\Delta S(T,p)}{nF}=V_{TN}(T,p)-\frac{T\Delta S(T,p)}{nF}$$
where *V_TN_*(*T*,*P*) is the minimum cell voltage under ideal conditions, defined as the thermoneutral voltage [31,32]. In standard conditions of temperature and pressure (*T* = 298.15 K and *p* = 1 bar), the changes of free energy, enthalpy, and entropy for the reaction given in Equation (6) in the decomposition of water are: [33,34,35].
$$\Delta G^{0}=237.13\;\mathrm{kJ}(i.e.\;E^{0}=\frac{\Delta G^{0}}{nF}=1.229\;V)$$
$$\Delta H^{0}=285.83\;\mathrm{kJ}\;(V^{0}=\frac{\Delta H^{0}}{nF}=1.481\;V)$$
$$\Delta S^{0}=163.15\;J\;{\mathrm{mol}}^{-1}K^{-1}\;(\frac{T\Delta S^{0}}{nF}=0.25\;V)$$

As is known, the potential that has to be applied to the electrolyzer to achieve the decomposition of water in the anode and the formation of hydrogen at the cathode for temperatures below 100 °C, assuming negligible concentration losses at the cathode, can be given by [36]
(9)$$E_{Cell}=E_{Nernst}+E_{Act}+E_{Ohm}$$
where
(10)$$E_{Nernst}=E^{0}+\frac{RT}{nF}\ln(\frac{p_{H_{2}}p_{O_{2}}^{0.5}}{1})$$

*E*^0^ is the reversible potential under standard conditions (*E*^0^ =1.23 V), supposing that $a_{H_{2}}=p_{H_{2}}$, $a_{O_{2}}=p_{O_{2}}$, and $a_{H_{2}O}=1$.

Considering the activation overpotential modeled by the Butler–Volmer equation, we can express
(11)$$E_{Act}=\frac{RT}{2\alpha F}\sinh^{-1}(\frac{j}{2j_{0,A}})+\frac{RT}{2\beta F}\sinh^{-1}(\frac{j}{2j_{0,C}})$$
where *j* is the current density, *j*_0,*A*_ and *j*_0,*C*_ are the exchange current density at the anode and cathode, respectively. *α* and *β* are the charge transfer coefficients at the anode and cathode, respectively.

Finally, the ohmic overpotential is
(12)$$E_{Ohmic}=\frac{j\cdot L}{\sigma }$$

*L* is the membrane thickness and membrane conductivity.

According to the first law of thermodynamics, the energy is conserved. Efficiency will, thus, be determined from the electrical energy converted into chemical energy. The thermodynamic efficiency of a PEM water electrolysis cell can be defined as
(13)$$\varepsilon =\frac{W_{t}}{W_{r}}$$
where *W_t_* is the theoretical amount of energy required to split one mole of water (*W_t_* = *V_TN_ I* t/n_H2_) and *W_r_* the real amount of energy to obtain it (*W_r_* = *E_cell_ I* t/n_H2_). Where *V_TN_* is the thermoneutral voltage and *E_cell_* the applied voltage to obtain it, *I* is the current intensity in the cell, *t* is the time, and *n* is the number of hydrogen moles. Efficiency can, thus, be expressed in terms of reversible voltage or in terms of thermoneutral voltage. Considering the thermoneutral voltage, we can express efficiency as
(14)$$\varepsilon =\frac{V_{TN}(T,p)}{E_{cell}(T,p,I)}$$

The Faradaic efficiency is defined as the ratio between the volume of hydrogen determined experimentally ($V_{H_{2}}(obtained)$ and the theoretically calculated ($V_{H_{2}}(calculated)$)
(15)$$\varepsilon_{Faradaic}=\frac{V_{H_{2}}(obtained)}{V_{H_{2}}(calculated)}$$
where the amount of hydrogen experimentally obtained and measured by the water–gas displacement method and that calculated are given by
(16)$$V_{H_{2}}(calculated)=\frac{R\cdot T}{p}\frac{I\cdot t}{2F}$$

Note that the external energy needed to produce one mole of hydrogen using methanol–water solution electrolysis under standard conditions will be around 43.8 kJ/mol instead of 286 kJ/mol for water electrolysis.

## 3. Results and Discussion

### 3.1. Results Obtained with Nafion 115 Membrane in Methanol—Water Solution Electrolysis

In the Nafion 115 water-electrolysis membrane, metallic platinum loadings of ca. 0.5 mg/cm^2^ are used as the cathode electrocatalyst to produce the hydrogen evolution reaction (HER), while 2 mg/cm^2^ of iridium oxide are used in the anode for the oxygen evolution reaction (OER). Although, the usual procedure implies the use of potentials higher than 0.5 V for methanol–water electrolysis and around 1.3 V in the case of pure water electrolysis, we worked using voltage experimental values until reaching a stable value. In this work, we have performed measurements at six different currents (0.02, 0.05, 0.1, 0.5, 0.8, and 1 A), obtaining a stable voltage in all cases. Typical polarization curves obtained at 30 °C in the 0 to 0.2 A/cm^2^ current density range are given in Figure 5 for different methanol concentrations (1, 2, 3, and 4 M) at atmospheric pressure (P = 1 bar). For a comparison, the behavior of pure water is also plotted. As inferred from the polarization curves, similar values were obtained for methanol–water electrolysis and pure water.

Figure 5 shows that the voltage drop is around 0.3 V at a current density of 0.15 A/cm^2^, when the methanol concentration changes from 1 to 2 M. This change increases up to 0.5 V, when the electrolysis of 1 M is compared with that of 3 M, and around 0.75 V when comparing 1 to 4 M. A comparison of the polarization curves between methanol 1 M electrolysis and pure water electrolysis shows a drop in voltage of about 400 mV. The reduction in the case of electrolysis for the different methanol concentrations are smaller at lower values of current density. At low current densities, the behavior changes due to the activation process; this process is more resistant when the solvent is pure water instead of methanol. The MEA resistance can be obtained from the slope of fitting in Figure 5 for the experimental values determined between 50 and 150 mA/cm^2^. A close inspection of these values shows that resistance is working less with methanol instead of pure water at current densities higher than 75 mA/cm^2^, where the variation of cell potential versus current density is practically linear due to the membrane resistance. The MEA resistance varied from 0.9 Ω when the solution was fed with methanol 4 M to 2 Ω when the solution was fed with pure water. Figure 5 shows that for cells operating with low methanol concentrations (1 to 3 M), the performance behaviors were very similar, but at concentrations of 4 M the activity dropped. It is worth mentioning that concentrations higher than 4 M were not tested due to possible dissolution of the Nafion ionomer present in the catalyst layer at high methanol concentrations [7]. This behavior could be related to the lower ion conductivity of the membrane when the methanol concentration is increased, as it swells, and proton diffusivity through the membrane decreases. Methanol crossover increases with methanol concentration and produces methanol vaporization as an impurity, together with hydrogen production at the cathode. For this reason, we chose 1 to 4 M as the optimal concentration range.

The energy consumption is about 0.21 kW·h/Nm^3^ of hydrogen at 1 A/cm^2^, at a methanol concentration of 4 M at 30 °C. This value is smaller than that observed for lower methanol concentrations (1, 2, and 3 M) in water electrolysis with the experimental setup. Our results in terms of energy consumption are lower than the values found using platinized titanium mesh on both sides of the catalyst in a Nafion 117 membrane as PEM, where the values were about 2.15 kW·h/Nm^3^ of hydrogen (at 0.5 A/cm2) and 2.87 kW·h/Nm^3^ of hydrogen (at 1 A/cm^2^) at the same temperature (30 °C) [37,38]. Our findings also show the suitability of aqueous methanol electrolysis, where at current densities higher than 0.5 A/cm^2^ the power consumption was 2.34 kW·h/kg of hydrogen, or about 18 times the electrical energy necessary to produce 1 kg of hydrogen by water electrolysis, which is estimated to be around 39.4 kW·h/kg or 3.54 kW·h/N m^3^ of hydrogen. In comparison, a commercial water electrolyzer has an energy consumption around 50–55 kW·h/kg, or 4.5–5 kW·h/N m^3^ of hydrogen [28]. 

Figure 6a shows the dependence of the hydrogen flow rate production of each electrolytic cell system on the current density in the electrolysis by methanol–water solution. This dependence was studied to verify the current efficiency of methanol electrolysis. For comparison, the same study was carried out using pure water in the electrolytic cell with the same Nafion 115 membrane. The flow rate of each methanol concentration in the electrolysis cells was calculated using the flow rate of the cathode. The hydrogen produced was measured by an AALBORG GFM 17 hydrogen flow meter, calibrated to reach a maximum value of 30 mL/min. As inferred from Figure 6a, the amount of theoretical hydrogen production measured at six different currents (0.02, 0.05, 0.1, 0.5, 0.8, and 1 A) has a linear behavior, with a slope of approximately 87 mL A^−1^ s^−1^ of the hydrogen produced in our cell, with the assumption that all of the current was used for hydrogen production. The hydrogen flow rate in the cathode increased in proportion to the current density, in agreement with the theoretical hydrogen production rate, as the slope is 32.5 L cm^2^ A^−1^ min^−1^. This shows that the hydrogen was effectively produced in the cathode, and the electrolytic cell performance was highly efficient, reaching values over 90% at 30 °C. Figure 6b displays the electrical energy needed to produce hydrogen in a methanol–water solution at different methanol concentrations (1, 2, 3, and 4 M) and in pure water, also measured at six different currents, i.e., 0.02, 0.05, 0.1, 0.5, 0.8, and 1 A. As shown, the experimental values for methanol–water electrolysis are quite similar to the theoretical value for water electrolysis. Thus, it can be concluded that methanol–water electrolysis needs considerably less electrical energy than water electrolysis to efficiently produce hydrogen.

### 3.2. Results Obtained by MF-4SC and MF-4SC/PANI Composite Membranes for Water Electrolysis

Proton transport in acidic membranes is a complex process involving the dissociation of the proton from the fixed sulfonic acidic group, its transference to the first hydration shell of water molecules, the separation of the hydrate proton from the conjugate base (anion of the acid group), and its diffusion, presumably stabilized as an Eigen-like cation in the confined water of the membrane matrix [39,40]. The large-scale connection of the water domains within the hydrated membrane and the flexibility of the skeletal bond of the polyelectrolyte chains also favor proton transport in the acidic membranes. One could, therefore, think that sulfonated fillers trapped in hydrophobic domains might connect with hydrophilic domains through hydrophobic channels, favoring proton transport. However, the conductivities measured by impedance spectroscopy for PANI nanocomposite membranes are, on average, about two–five times higher than those of pristine MF-4SC membranes at temperatures of 50–80 °C. To explain this, we hypothesized that the PANI fillers in the membranes presumably create percolation paths, through which additional proton diffusion takes place, increasing conductivity. In a previously published paper [41,42,43,44], we observed that by increasing polymerization time by approximately 30 days, membrane saturation by polyaniline reaches a limiting value. This could be an indication of morphological changes in the cluster zones of the composite membrane due to the transition from pristine MF-4SC to nanosized polyaniline clusters in MF-4SC/PANI-3H* and MF-4SC7/PANI-3H** [45]. These nanosized domains, formed by a mixture of pernigraniline and emeraldine, can then produce growing percolation paths in which the proton moves more easily. These nanosize domains do not seem to disturb the already-existing percolation paths in the pristine MF-4SC membranes.

The SEM images of the surface and cross section (Figure 1) of both the pristine MF-4SC and the MF-4SC/PANI-3H* nanocomposite membranes may be an indication of the MF-4SC/PANI-3H* anisotropy after the template synthesis. The MF-4SC/PANI-3H* membrane surface shows a grainy, filamentous aspect, almost absent in the MF-4SC pristine membrane. This filamentous surface aspect of the MF-4SC/PANI-3H* membrane can vary from filamentous to grainy, depending on the supporting acid used in its synthesis [43,45], which is due to the deposition of PANI on the membrane surface. The real size of the grains is approximately 1–5 µm, with a fractal dimension of about 2.5, which is close to the spherical geometry of the surface particles. The MF-4SC/PANI-3H* membrane shows some aggregates along the surface (Figure 1d) and in the samples (Figure 1d), forming some structural cavities. A portion of these cavities in the membrane could reach from one side to the other, thus favoring the growth of the PANI chains through the whole membrane. The filamentous, grainy structure is not only present on the membrane surface but also in its bulk. Similar results have been confirmed by Berezina et al. for PANI membranes from elemental analysis [30,43,45].

The typical polarization curves obtained using different membranes (MF-4SC, MF-4SC/PANI*, and MF-4SC/PANI**) sandwiched between the same electrodes as in the study, carried out on methanol–water solution electrolysis, but here in a water-electrolysis cell at different temperatures (30, 50, and 80 °C), are given in Figure 7. The current density range was 0–2 A/cm^2^. The electrochemical performances measured using the perfluorinated sulfocationic MF-4SC membrane and nanocomposites of MF-4SC/PANI membranes produce efficient results when the cell voltage decreases as temperature increases, which brings the cell voltage closer to the higher heating-value voltage, resulting in high electrical efficiency. 

As can be seen in Figure 7, the actual cell voltage is higher than the highest heating value voltage at low temperatures, resulting in low electrical energy efficiency. This confirms that the water electrolyzer performs better at higher temperatures, with better performance by the composite MF-4SC/PANI membranes than the MF-4SC membranes. This could be related to the membrane properties and can be achieved by reducing the ohmic resistance, using thinner membranes, and reducing activation losses. Figure 7 shows that membrane conductivity can be obtained by determining the resistance of the MEA from the slope of the straight line in the range of current intensity between 0.75 and 2 A/cm^2^, which is that of a cell voltage between 2.25 and 3.15 V. The values calculated for all membranes and temperatures are given in Table 2. As inferred from the values, the conductivity increases as a function of temperature, as reported in previous works [40,41].

To compare the calculated conductivity values in Figure 7, we also determined membrane conductivity from the electrochemical impedance spectroscopy (EIS) measurements in the frequency range interval 0.1 Hz to 10^7^ Hz at 30 °C, following the same procedure described previously [45,46,47,48,49,50,51]. Although there are different procedures available to obtain dc conductivity, all the methods are indirect because they lack the criteria to give the sample dc conductivity value. One consists of using the Nyquist plot, in which the imaginary part versus the real parts of the impedance is plotted (-Z′′ versus Z′) [47,48,49,50,51]. Following this procedure, the Nyquist plots of the different samples are measured at 30 °C, as depicted in Figure 8. The real part of the impedance was obtained from the inset plot. This value can be assigned to the membrane sandwiched between the two electrodes, where the values of 0.47, 0.54, and 0.63 Ω were determined for the samples MF-4SC, MF-4SC-PANI*, and MF-4SC-PANI**, respectively. Knowing the thickness and the area of the membrane (i.e., the area of the electrolyzer cell) in contact with the electrodes, from these values we obtained the values 5.3 × 10^−3^, 8.1 × 10^−3^, and 7.0 × 10^−3^ S/cm for the conductivity of MF-4 SC, MF-4SC-PANI*, and MF-4SC-PANI**, respectively. These values are more than one order of magnitude higher than the values of the polarization curves of MF-4SC, MF-4SC-PANI*, and MF-4SC-PANI**, respectively, determined from Figure 6b, which were found to be 0.125, 0.135, and 0.22 S/cm.

Figure 9 shows that the efficiency of a PEM water electrolysis cell is one of the more critical parameters for characterizing the energy cost of the electrocatalytical process. As can be seen in Figure 9, the efficiency drops as the current intensity increases, so that a compromise has to be found when comparing cost with energy. In our study, when MF-4SC is the PEM water electrolyzer, the average efficiency is around 42% at 80 °C, and *p* = 1 bar. This value is below 47% for MF-4SC/PANI* composite membranes and 49% for MF-4SC/PANI** composite membranes measured at 1 A/cm^2^ current density, while the average efficiency measured in the complete range of current intensities was around 70% for all membranes.

Given the fact that the variation of the efficiency with the current density for nanocomposite membranes is almost the same over the entire current efficiency range at all the temperatures studied, it can be concluded that the composites display better behavior in terms of efficiency than the pristine MF-4SC membrane.

Figure 10 shows that the specific energy needed to produce hydrogen by water electrolysis is dependent on temperature. For example, at 30 °C, membrane MF-4SC/PANI** needs less specific energy than either MF-4SC/PANI* or MF-4SC. However, at 50 °C, all the membranes have similar behavior, and when the temperature is raised to 80 °C, less specific energy is required using membrane MF-4SC/PANI*, while both composite PAni membranes present better behavior than MF-4SC at around a 20% reduction in specific energy. The enhanced performance of nanocomposite PANI membranes is attributed to improve water retention, because the PAni fillers in the MF-4SC membranes presumably cause percolation paths through which additional proton diffusion takes place, increasing proton conductivity; so, as a consequence, more hydrogen will be generated in the cathode.

The present findings, therefore, confirm that using methanol as the hydrogen source is highly beneficial, since much less energy is required for its generation. The decomposition of methanol to hydrogen in the electrolyzers and using hydrogen in a conventional PEM cell, which is much more active than a DMFC cell, are, therefore, interesting from an industrial point of view as an energy innovation. The main advantage of methanol electrolyzers is the substantial energy savings that they provide over water electrolyzers, since only potentials from 0.4 to 1 V per cell are required, while pure water electrolyzers require potentials of the order of 1.4 V, depending on the working temperature. The savings achieved are, thus, so significant that even producing hydrogen by methanol electrolysis costs less than using water, even when taking the cost of the methanol into account.

## 4. Conclusions

In conclusion, we have shown that, by coupling methanol electrolyzers to a hydrogen storage tank and a PEM cell, it would be possible to achieve higher energy storage yields than if we consider the electricity applied to the electrolyzers and that from the fuel cell, since the potentials and current densities per electrolyzer cell and fuel cell may overlap. We hypothesized that the use of PANI fillers in the membrane presumably creates percolation pathways, allowing additional proton diffusion to take place, which increases conductivity. Surface and cross-section SEM images of PANI-3H**/MF-4SC exhibit a filamentous grainy structure, not only on the membrane surface but also in its bulk. This filamentous structure is almost completely absent in the images obtained from the pristine MF-4SC membrane. The MEA conductivity values measured by impedance spectroscopy are in fair agreement with those obtained from the ohmic regime of the polarization curves and are close to the Nafion 117 values, as reported in the literature. The obtained efficiency, using as electrocatalyst in the cathode, Pt (30 wt%) on Vulcan carbon powder XC72 with a load of 0.5 mg/cm^2^ on the membrane surface, and 2 mg/cm^2^ of iridium oxide (IrO_2_) on the surface as anode, is, in general, about 60% for all the membranes operating at 1 A/cm^2^. Our results show that PEM electrolyzers provide a hydrogen production of 30 mL/min working at 160 mA/cm^2^. As shown, composite PANI membranes present better behavior than pristine perfluorinated sulfocationic membranes, which could presumably be attributed to the generation of percolation pathways through which additional proton diffusion might occur, increasing the proton conductivity and, consequently, favoring the generation of hydrogen in the cathode. On the other hand, water retention will also be produced, and a reduction in specific energy of around 20% is observed. 

The electrical energy needed to produce hydrogen by methanol–water electrolysis was found to be much less than by water electrolysis, around 65% of that required by water electrolysis. The hydrogen flow rate in the cathode increased in proportion to the current density and almost agreed with the theoretical hydrogen production rate. Finally, with methanol electrolysis at current densities higher than 0.5 A/cm^2^, the power consumption is 2.34 kWh/kg of hydrogen. This value is about 18 times lower than the electrical energy necessary to produce 1 kg of hydrogen by water electrolysis, which is estimated to be around 39.4 kW h/kg or 3.54 kW h/N m^3^ of hydrogen, while a commercial water electrolyze needs around 50–55 kW h/kg or 4.5–5 kW h/N m^3^ of hydrogen. These results, showing the superior power consumption of the Nafion/PANI membrane, might pave the way toward developing future PEMWE cells with superior performance.

## Figures and Tables

**Figure 1 polymers-14-04500-f001:**
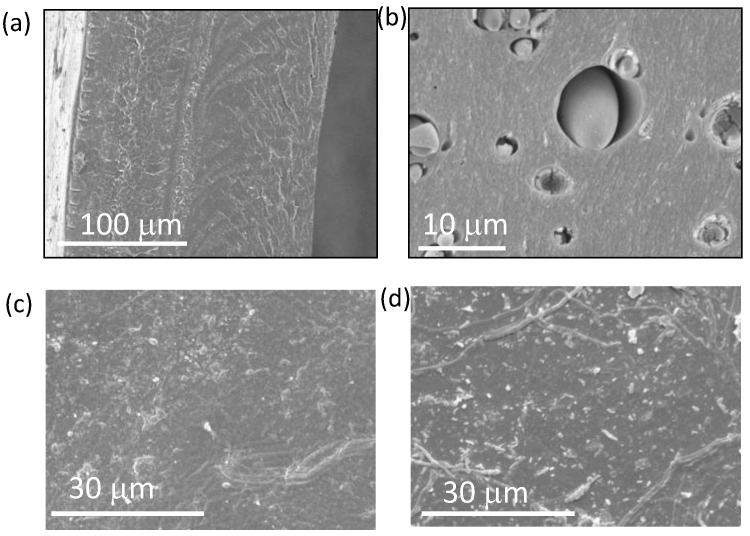
SEM images of the surface of the MF-4SC membrane (**a**) and composite membrane MF-4SC/PANI-3H* (**b**). SEM images of the cross section of an MF-4SC membrane (**c**) and composite membrane MF-4SC/PANI-3H* (**d**).

**Figure 2 polymers-14-04500-f002:**
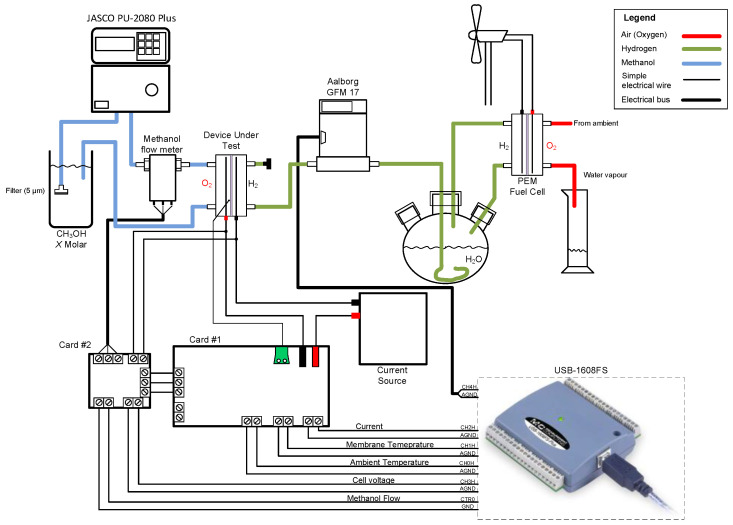
Diagram of experimental setup.

**Figure 3 polymers-14-04500-f003:**
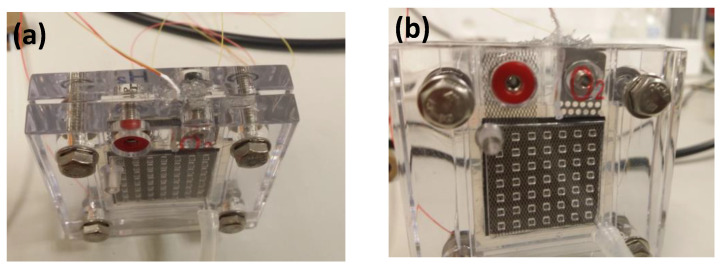
Image of the single cell with the device under test and the thermocouple insertion (**a**) Top view, (**b**) Front view.

**Figure 4 polymers-14-04500-f004:**
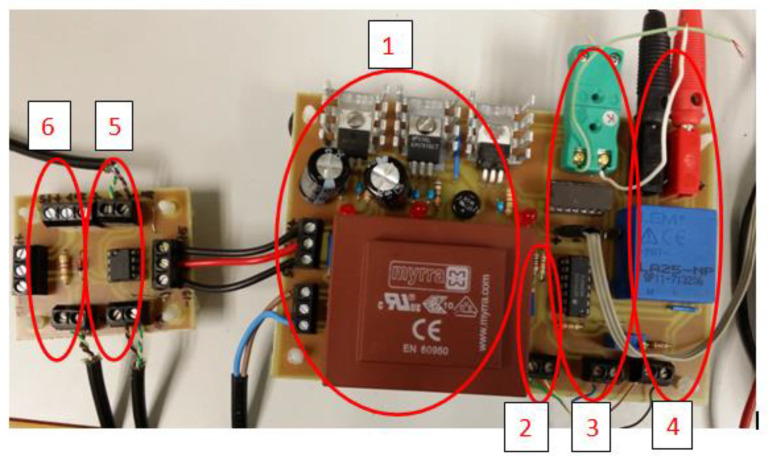
Image of the two interface cards. 1, power supply; 2, conditioning of the ambient temperature sensor; 3, conditioning of the cell temperature sensor; 4, conditioning of the current measuring; 5, cell voltage conditioning; 6, conditioning of pulses from the methanol flow meter.

**Figure 5 polymers-14-04500-f005:**
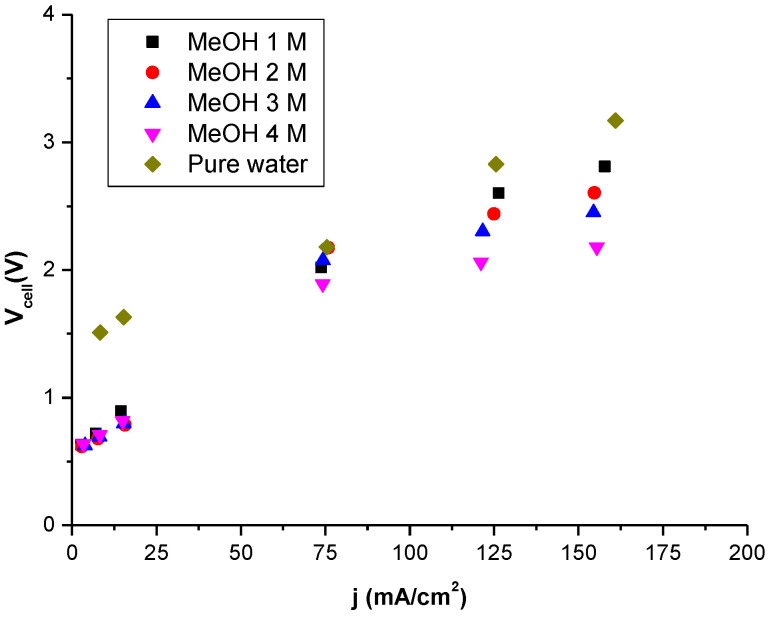
Current density-voltage performances measured in a single cell with a Nafion 115 membrane at different methanol (MeOH) concentrations (1, 2, 3, and 4 M) and pure water. Operating conditions: T_cell_ = 30 °C, and P = 1 bar.

**Figure 6 polymers-14-04500-f006:**
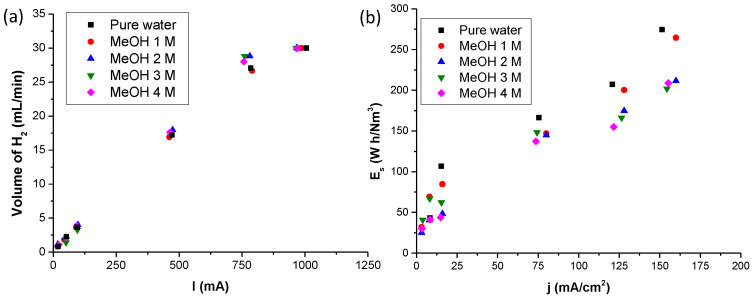
(**a**) Volume of hydrogen obtained versus provided current for each methanol (MeOH) concentration and pure water solution. (**b**) Comparison of electrical energy needed to produce hydrogen in electrolysis by methanol–water solution at different methanol (MeOH) concentrations and in pure water.

**Figure 7 polymers-14-04500-f007:**
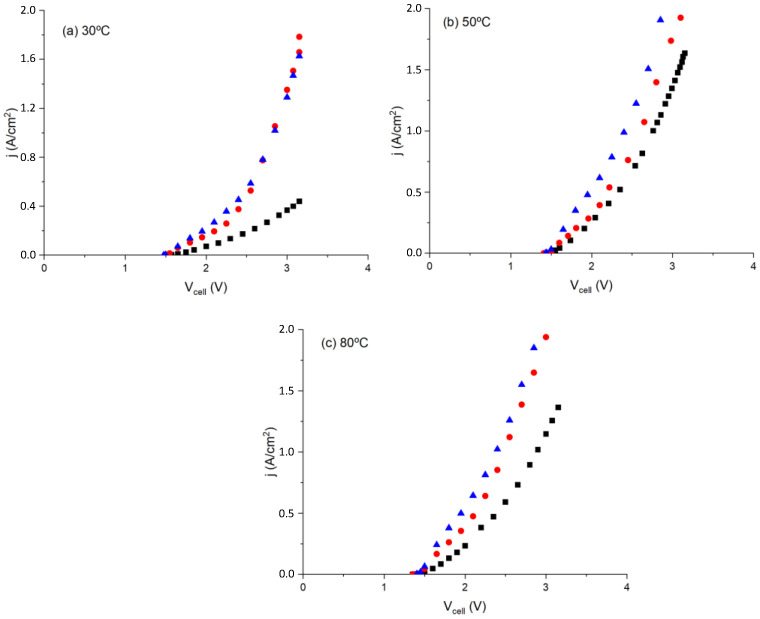
Polarization curves of MF-4SC (■), MF-4SC-PANI* (
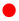
), and MF-4SC-PANI** (
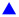
) membranes at different temperatures.

**Figure 8 polymers-14-04500-f008:**
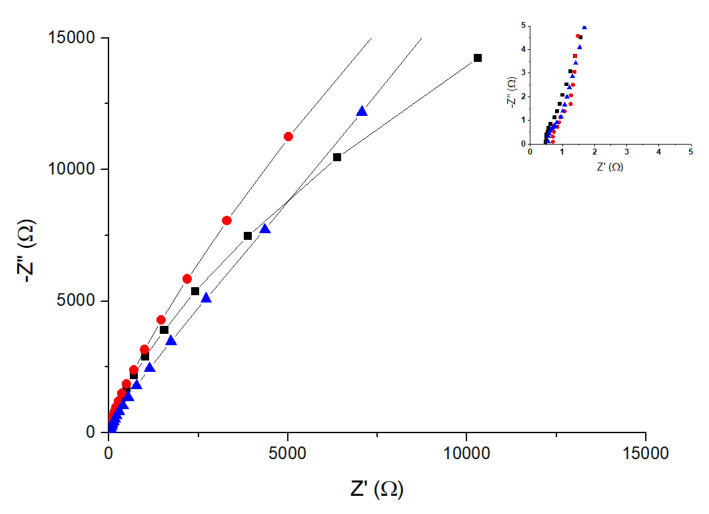
Nyquist diagram obtained for the samples MF-4SC (■), MF-4SC-PANI* (
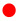
), and MF-4SC-PANI** (
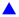
) at 30 °C. The inset shows the intercept at the real part of the impedance axes, to indicate membrane resistance.

**Figure 9 polymers-14-04500-f009:**
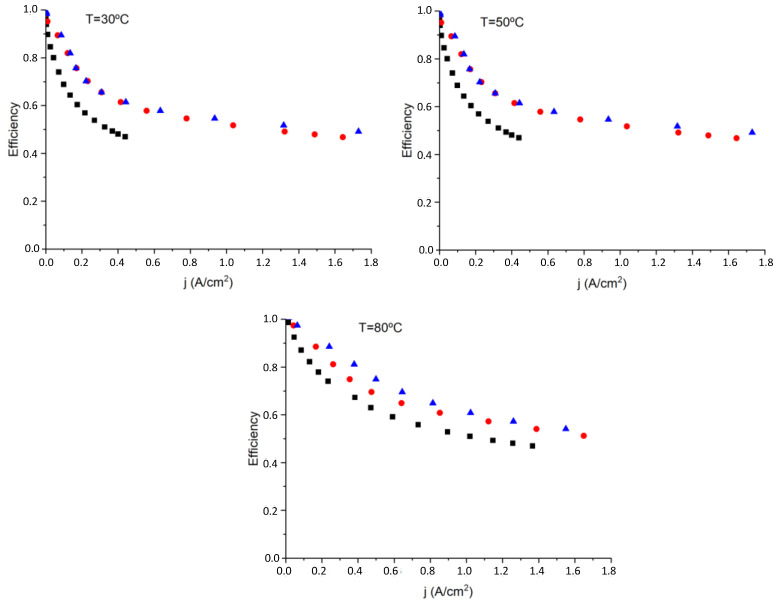
Efficiency variation obtained from thermoneutral voltage of the membranes MF-4SC (■), MF-4SC-PANI* (
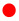
), and MF-4SC-PANI** (
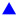
), respectively, at different temperatures: 30, 50, and 80 °C, respectively.

**Figure 10 polymers-14-04500-f010:**
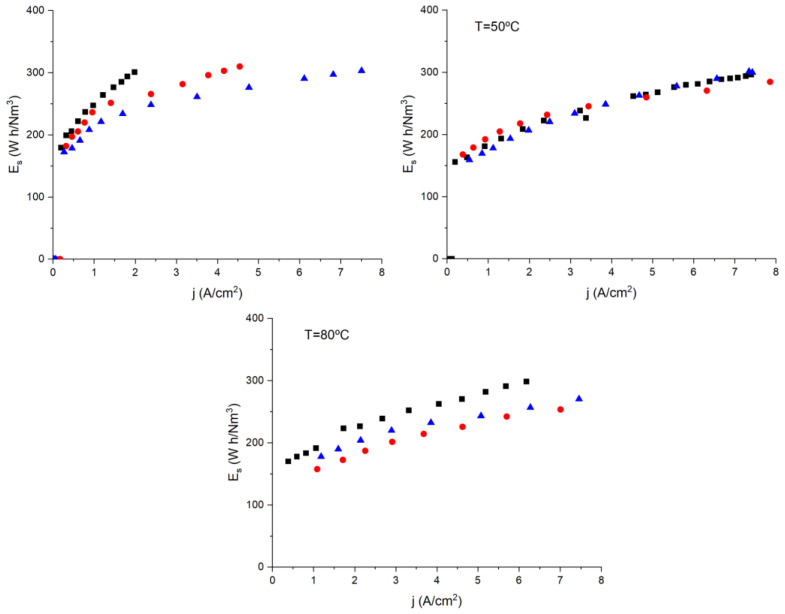
Specific energy consumption required to produce hydrogen by water electrolysis for the different membranes MF-4SC (■), MF-4SC-PANI* (
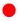
), and MF-4SC-PANI** (
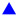
), respectively, at: 30, 50, and 80 °C, respectively.

**Table 1 polymers-14-04500-t001:** Thickness, saturation degree expressed as content of polyaniline (%), water uptake, ion-exchange capacity (IEC) of Nafion and perfluorinated sulfocationic MF-4SC, MF-4SC/PANI-3H*, and MF-4SC/PANI-3H** nanocomposite membranes.

Membrane	Thickness (μm)	Saturation Degree (%)	Water Uptake	IEC (Equiv. H^+^/g Wet Membrane)
MF-4SC	255 ± 5	----	23.6 ± 0.3	0.70 ± 0.01
MF-4SC/PANI-3H*	274 ± 5	10.4	22.5 ± 0.3	0.69 ± 0.01
MF-4SC/PANI-3H**	275 ± 5	10.6	22.6 ± 0.3	0.69 ± 0.01
Nafion 115	155 ± 2	----	35.2 ± 0.5	1.01 ± 0.01

**Table 2 polymers-14-04500-t002:** Conductivity values of the MEA determined from the fitting of the polarization curves shown in Figure 8 at different temperatures.

Membrane	σ (S/cm) at 30 °C	σ (S/cm) at 50 °C	σ (S/cm) at 80 °C
MF-4SC	0.009	0.025	0.075
MF-4SC-PANI*	0.135	0.210	0.230
MF-4SC-PANI**	0.220	0.126	0.220

## Data Availability

Data presented in this study are available upon request from the corresponding author.

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
