# Peer review of "Hydrogen Production from Methanol–Water Solution and Pure Water Electrolysis Using Nanocomposite Perfluorinated Sulfocationic Membranes Modified by Polyaniline"

_polymers, 2022, doi:10.3390/polym14214500_

Round 1
Reviewer 1 Report
The manuscript must be improved before the publications because in the current form is not suitable for the publication, despite the topic is relevant for scientific community
More comments are reported below:
GENERAL
-If a further revision should be needed please add the line numbers
-Please check carefully the english. The paper is hard to read in the present form. I suggest to make shorter sentences.
ABSTRACT
- "Operating conditions are optimized in terms of current density, stability and meth-anol concentration. The effect of various parameters such as methanol concentration, water and cell voltage on cell performance was investigated". These two sentences are rendundant please adjust the text.
-"much less than that in water electrolysis and around 65% of that in the water electrolysis." this sentence is not clear, please rewrite it.
-"The results shown..." the form of this sentence is not linear and quite hard to follow. In particualr the beginning "The results shown the main characteristics.." is not so appropriate for a scientific paper.
INTRODUCTION
- "There are four types of processes to produce hydrogen via electrochemical:..." Please rewrite this sentence
- Ref [2-10] are used as references for H2 production pathways. All the papers have more than 10 years, then, considering the strong efforts and fast development I suggest to update a little bit the references adopted here. Not all, but to show that during the last years important improvments have been provided.
- Also the following sentence "These methods have their advantages,..." is not acceptable in the current form.
- By the introduction is really hasrd to understand the aim of the work which is focused on membrane modification. Why is so important such modification for cell performance?
2.3 MICROSCOPY MEASUREMENTS
- Figure 1 reports results of the experiments carried out so it should be move to the right section. Moreover I advice to add letters from a to make the reading easier.
- "Methanol or pure water flow. Range: 0,005 – 1,75 l/min" please substitute the comma with the dot for decimal digit
3.1. METHANOL-WATER SOLUTION ELECTROLYSIS
- Correct the formula o methanol in equation 1 and 3
4. RESULTS
- Why the polarization curves of figure 5 have been collected with a so strange sampling rate and not with a constant step?
-Please use in Fig.7 the same symbols used in Fig. 6 and 5
-"Take et al [28] has reported similar" the reference [28] is different (Aldebert et al). Take is [27]
Author Response
Reviewer1
GENERAL
-If a further revision should be needed please add the line numbers
REPLY: Thanks to the reviewer. In the new version of the manuscript we have added the line numbers in each file of the manuscript.
-Please check carefully the english. The paper is hard to read in the present form. I suggest to make shorter sentences.
REPLY: Thanks to the reviewer. In the new version of the manuscript the English has been noticeably corrected as it has been rewritten by an English-speaking proofreader.
ABSTRACT
- "Operating conditions are optimized in terms of current density, stability and meth-anol concentration. The effect of various parameters such as methanol concentration, water and cell voltage on cell performance was investigated". These two sentences are rendundant please adjust the text.
-"much less than that in water electrolysis and around 65% of that in the water electrolysis." this sentence is not clear, please rewrite it.
-"The results shown..." the form of this sentence is not linear and quite hard to follow. In particular the beginning "The results shown the main characteristics.." is not so appropriate for a scientific paper.
REPLY: Many thanks to the reviewer. In the revised manuscript all the sentences have been corrected.
Abstract: In this work we synthesized a Nafion membrane and two different nanocomposite MF-4SC membranes modified by Polyaniline by casting using two different polyaniline infiltration procedures. Operating conditions were optimized in terms of current density, stability and methanol concentration. A study was made of the effects on cell performance of various parameters, such as methanol concentration, water and cell voltage. The energy required for pure water electrolysis was analyzed at different temperatures for all three membranes. Our experiments showed that PEM electrolyzers provide a hydrogen production of 30 mL/min working at 160mA/cm2. Our composite PAni membranes showed better behavior than pristine perfluorinated sulfocationic membranes (around 20% reduction in specific energy). Methanol-water electrolysis required considerably less (around 65%) electrical power than water electrolysis. The results provided the main characteristics of aqueous methanol electrolysis, in which the power consumption is 2.34 kWh/kg of hydrogen at current densities higher than 0.5A/cm2. This value is about eighteen times less than the electrical energy required to produce 1 kg of hydrogen by water electrolysis, estimated to be around 39.4 kW h/kg or 3.54 kWh/N m3 of hydrogen, while for a commercial water electrolyser it is about 50–55 kWh/kg or 4.5–5 kWh/N m3 of hydrogen.
INTRODUCTION
- "There are four types of processes to produce hydrogen via electrochemical:..." Please rewrite this sentence
REPLY: Many thanks to the reviewer. The sentence has been rewrite in the new version of the manuscript.
….“Hydrogen can be produced by five processes: electrochemical, thermochemical, photochemical, biological and electrolysis [2-10].”
- Ref [2-10] are used as references for H2 production pathways. All the papers have more than 10 years, then, considering the strong efforts and fast development I suggest to update a little bit the references adopted here. Not all, but to show that during the last years important improvments have been provided.
REPLY: For the authors the references used from 2 to 10 correspond to basic papers used in the beginning of this century, basically between 2003 and 2010. We think that they are pioneers in the study of water and methanol-water electrolysis. Nevertheless, in the new version we have added the references 16, 17, 18, 19 and 20 published more recently. We think that these references can help to improve our manuscript in agreement with the suggestion of the reviewer1.
- Also the following sentence "These methods have their advantages,..." is not acceptable in the current form.
REPLY: Thanks to the reviewer. In the new version we have modified to: ……Each method has its advantages and disadvantages but the most eco-friendly method of pure hydrogen production on a small scale is PEM water electrolysis.
- By the introduction is really hasrd to understand the aim of the work which is focused on membrane modification. Why is so important such modification for cell performance?
2.3 MICROSCOPY MEASUREMENTS
- Figure 1 reports results of the experiments carried out so it should be move to the right section. Moreover I advice to add letters from a to make the reading easier.
REPLY: The Figure 1 have been moved to the results section.
- "Methanol or pure water flow. Range: 0,005 – 1,75 l/min" please substitute the comma with the dot for decimal digit
REPLY:: In the new version the comma have been changed by decimal digit
3.1. METHANOL-WATER SOLUTION ELECTROLYSIS
- Correct the formula o methanol in equation 1 and 3
REPLY: Thank you very much to reviewer. In the new manuscript the mistake have been corrected.
RESULTS
- Why the polarization curves of figure 5 have been collected with a so strange sampling rate and not with a constant step?
REPLY: Many thanks to the reviewer. One error in the file of the measurements with water was found and we have corrected the mistake in Figure 5 of the new version of the manuscript.
Although, the usual procedure would have to choose potentials greater than 0.5 V for metahnol-warter electrolysis and around 1.3 V in case of pure water electrolysis, respectively, increasing the voltage untilto reach a current density stable, we have worked fixing the current measuring the voltage experimental values until it reach a stable value. In this work we have measured at six different currents (0.02A, 0.05A, 0.1A, 0.5A, 0.8A and 1A, respectively), in all cases the stabilized voltage. Typical polarization curves obtained at 30ºC in the 0 to 0.2 A/cm2 current density range are given in Figure 5 for different methanol concentrations (1M, 2M, 3M and 4M). The behavior compared to that of pure water is also plotted for comparison. Both measurements were made at atmospheric pressure (p=1bar). From this figure we can be seen that the experimental values for methanol-water electrolysis and pure water have a similar behavior, but at very low current density the tendency to the theoretical voltage value for water electrolysis and methanol-water solutions was observed.
Figure 5. Current density-Voltage performances measured in a single cell with a Nafion®115 membrane at different methanol concentrations (1M, 2M, 3M and 4M), operating conditions: Tcell=30ºC and p = 1bar. The performance measured using pure water is also plotted for comparison.
A close inspection of Figure 5 shows that the voltage drop is around 0.3V at a current density of 0.15A/cm2, when the methanol concentration changes from 1M to 2M. This change is 0.5 V when we compare the electrolysis with 1M respect 3M, and around 0.75V comparing 1M to 4M. When we compare the polarization curves between methanol water 1M electrolysis and pure water electrolysis we observe a drop in voltage about of 400mV.
-Please use in Fig.7 the same symbols used in Fig. 6 and 5
REPLY: Thank you very much to reviewer. In the new manuscript the symbols have been changed using the same symbols y all figures 5, 6 and 7.
-"Take et al [28] has reported similar" the reference [28] is different (Aldebert et al). Take is [27]
REPLY: Thank to the reviewer. One mistake in the numeration of our references have been corrected in the new version.
Reviewer 2 Report
Comments and Suggestions for Authors
Dear Authors,
The Title:
Hydrogen Production From Methanol-Water Solution and Pure Water Electrolysis Using Nanocomposite Perfluorinated Sulfocationic Membranes Modified by Polyaniline
I have to read your manuscript with great attention and interest.
The authors of the study investigated the effects of polyanoline synthesized in two ways and used for the production of hydrogen. Composite diaphragm has been shown to perform better than pure Nafion. The research is compact and consistent. The research methodology was correctly applied and the conclusions from the research were formulated.
The submission falls within the scope of the journal and is sufficiently original.
Detailed comments are provided below:
- Equation 1,3 – methanol – CH3OH
- use unit mol dm-3, instead symbol “M”
- Fig. 6, Fig. 7,- cut the axes to get better quality
- Improve tha quality of Fig. 8, inset in Fig. 9, Fig. 10, Fig. 11
- prepare the conclusion in point
- can you give us the mechanism of catalytic action of polyaniline using two different procedures?
Author Response
Reviewer2
I have to read your manuscript with great attention and interest.
The authors of the study investigated the effects of polyanoline synthesized in two ways and used for the production of hydrogen. Composite diaphragm has been shown to perform better than pure Nafion. The research is compact and consistent. The research methodology was correctly applied and the conclusions from the research were formulated.
The submission falls within the scope of the journal and is sufficiently original.
Detailed comments are provided below:
- Equation 1,3 – methanol – CH3OH
REPLY:Thank you very much to reviewer. In the new manuscript the mistake has been corrected.
- use unit mol dm-3, instead symbol “M”
REPLY: In the new version has been modified.
- Fig. 6, Fig. 7,- cut the axes to get better quality
REPLY: In the new version has been modified.
- Improve the quality of Fig. 8, inset in Fig. 9, Fig. 10, Fig. 11
REPLY:In the new version the figures has been improved.
- prepare the conclusion in point
REPLY: The conclusion in point has been prepared taking into account the change done.
- can you give us the mechanism of catalytic action of polyaniline using two different procedures?
REPLY: Is known than PAni nanocomposite membranes, the proton conductivities are around 3 to 5 times higher than the corresponding to the pristine MF-4SC membranes at low temperatures (between 50 to 80ºC). See references: Yang et al. J. Power Sources, 189,1016 (2009) and A. Munar et al. Journal The Electrochemical Society, 157 (8) B1186-B1194 (2010). The enhanced performance of nanocomposite PAni membranes is attributed to improve water retention, because the PAni fillers in the MF-4SC membranes presumably cause percolation paths through which additional proton diffusion through the polymeric membrane takes place increasing its conductivity and as consequence more hydrogen is generated in the cathode.
Reviewer 3 Report
The manuscript doesn't qualify for publishing in any journal in the present format with a lot of typos, grammatical error, scientific error. It sounds like authors used some online tools to transalate to English language. Authors are encouraged to do thorough revision.
Author Response
Reviewer3
The manuscript doesn't qualify for publishing in any journal in the present format with a lot of typos, grammatical error, scientific error. It sounds like authors used some online tools to transalate to English language. Authors are encouraged to do thorough revision.
REPLY: We thank reviewer3 for the revision of our manuscript. In the revised manuscript in agreement with the reviewer’s comments we think that grammatical and scientific mistakes have been corrected. The English language has been improved in the new version. The authors think that the reviewer3 can verify that the work has been improved and we think that his impression on the publication of the article will have changed.
Round 2
Reviewer 1 Report
The Authors reply in a sufficient way to all points of the previous revision, so the paper now is in a form that can be accept for the publication
Author Response
Many thanks to the reviewer1 .
Reviewer 2 Report
The authors made changes in accordance with the comments of the reviewer.
Author Response
Many thanks to the reviewer2 .
Reviewer 3 Report
The manuscript has improved significantly after revision. A few specific comments are:
1. The introduction doesn't provide any clue on the overall focus of the article. Line 80-89, authors describe H2 production by two different methods, but there is no clear description on the significance of the present work. Why do you need to make composite membrane ? What is the expected role of PAni ? Are there any previous reports on similar systems? If so, how the present work is different ?
2. Line 49-50: The oxygen evolution reaction occurs at the anode with
iridium dioxide (IrO2) as the typical catalyst, due to the higher electrode potential (above 1.48 V) than the standard hydrogen electrode. What is electrode potential here ?
3. Line 59-60: Electricity is known to be the most expensive form of energy (237.2 kJ/mol). Why is the relevance of free energy value here ?
4. Line 61-62: However, water electrolysis at present only produces 4% of global industrial hydrogen. Remove however.
Author Response
Reviewer3 comments:
The manuscript has improved significantly after revision. A few specific comments are:
REPLY: Many thanks to reviewer by consider that the manuscript have been improved.
- The introduction doesn't provide any clue on the overall focus of the article. Line 80-89, authors describe H2 production by two different methods, but there is no clear description on the significance of the present work. Why do you need to make composite membrane ? What is the expected role of PAni? Are there any previous reports on similar systems? If so, how the present work is different ?
REPLY: Think you very much by the suggestion given by the reviewer. In the new version of the introduction we think that manuscript provide a clear description of our work. More refernces have been added.
- Line 49-50: The oxygen evolution reaction occurs at the anode with
iridium dioxide (IrO2) as the typical catalyst, due to the higherelectrode potential (above 48 V) than the standard hydrogen electrode. What is electrode potential here ?
REPLY: The paragraph has been changed.
- Line 59-60: Electricity is known to be themost expensive form of energy (237.2 kJ/mol). Why is the relevance of free energy value here ?
REPLY: The paragraph has been changed
- Line 61-62: However,waterelectrolysis at present only produces 4% of global industrial hydrogen. Remove however.
REPLY: Thanks to the reviewer. In the new version the sentence is changed.
Sincerely yours,
Vicente Compañ
Round 3
Reviewer 3 Report
The authors are repeating typos and grammar mistakes even in the revised version. I wouldn't recommend for publication. Publishing such a low quality article may affect the reputation of the journal. I also do not want to review this article anymore. Also, I would like not to be mentioned as the reviewer for this article anywhere.
Author Response
Reviewer3 comments:
The authors are repeating typos and grammar mistakes even in the revised version. I wouldn't recommend for publication. Publishing such a low quality article may affect the reputation of the journal. I also do not want to review this article anymore. Also, I would like not to be mentioned as the reviewer for this article anywhere.
REPLY: Think you very much by the suggestion given by the reviewer. With the aim of considering the suggestions made by reviewer 3, the work has been modified and the English grammatically has been corrected by a native speaker.
In the new version of the introduction we think that manuscript provide a clear description of our work. The manuscript has been rewritten clearly indicating everything that has been done. Some figures have also been regrouped. More references have been added.
Sincerely yours,
Vicente Compañ